# Evaluating the Need for Pre-CoMiSS™, a Parent-Specific Cow’s Milk-Related Symptom Score: A Qualitative Study

**DOI:** 10.3390/nu17091563

**Published:** 2025-04-30

**Authors:** Yvan Vandenplas, Kateřina Bajerová, Christophe Dupont, Mikael Kuitunen, Rosan Meyer, Anna Nowak-Wegrzyn, Carmen Ribes-Koninckx, Silvia Salvatore, Raanan Shamir, Annamaria Staiano, Hania Szajewska, Carina Venter, Sue Jones, Anette Järvi, Catherine Couchepin

**Affiliations:** 1KidZ Health Castle, UZ Brussel, Vrije Universiteit Brussel, 1090 Brussels, Belgium; 2Department of Pediatrics, University Hospital Brno, Faculty of Medicine, Masaryk University, 62500 Brno, Czech Republic; bajerova.katerina@fnbrno.cz; 3Clinique Marcel Sembat, Ramsay Group, 92100 Paris, France; christophe.dupont@wanadoo.fr; 4Children’s Hospital, Helsinki University Hospital, University of Helsinki, 00014 Helsinki, Finland; mikael.kuitunen@helsinki.fi; 5Faculty of Medicine, KU Leuven Belgium, 3000 Leuven, Belgium; rosan.research@rosan-paediatricdietitian.com; 6Hassenfeld Children’s Hospital, Department of Pediatrics, NYU Grossman School of Medicine, New York, NY 10016, USA; anowak_wegrzyn@hotmail.com; 7Department of Pediatrics, Gastroenterology and Nutrition, Collegium Medicum, University of Warmia and Mazury, 10719 Olsztyn, Poland; 8Pediatric Gastroenterology Unit, La Fe Hospital Research Institute, 46026 Valencia, Spain; carmen_ribes@iislafe.es; 9Department of Medicine and Technological Innovation, Pediatric Unit, University of Insubria, 21100 Varese, Italy; silvia.salvatore@uninsubria.it; 10Institute of Gastroenterology, Nutrition and Liver Diseases, Schneider Children’s Medical Center of Israel, Petah Tikva 4920235, Israel; raanan@shamirmd.com; 11School of Medicine, Faculty of Medical and Health Sciences, Tel Aviv University, Tel Aviv 6997801, Israel; 12Department of Translational Medical Sciences, University of Naples “Federico II”, 80138 Naples, Italy; staiano@unina.it; 13Department of Paediatrics, Medical University of Warsaw, 02-091 Warsaw, Poland; szajewska@gmail.com; 14Section of Allergy and Clinical Immunology, Children’s Hospital Colorado, University of Colorado, Aurora, CO 80045, USA; carina.venter@childrenscolorado.org; 15Nestlé Health Science, 1800 Vevey, Switzerland; susan.jones@nestle.com (S.J.); anette.jarvi@nestle.com (A.J.); 16Ipsos SA, 1219 Geneva, Switzerland; catherine.couchepin@ipsos.com

**Keywords:** cow’s milk, cow’s milk allergy, crying, distress, food allergy, infant, regurgitation, CoMiSS™

## Abstract

**Background/Objectives**: Cow’s milk allergy (CMA) presents a significant clinical burden. The Cow’s Milk-related Symptom Score (CoMiSS™) is a widely used clinical screening tool designed to raise awareness of CMA among healthcare professionals. This qualitative study aimed to assess the need for a parent-reported CoMiSS™ tool (Pre-CoMiSS™) and explore its potential usefulness for parents and primary care physicians (PCPs). **Methods**: Participants were parents of infants aged 2–12 months and PCPs from Germany, Sweden, Spain, and the United Kingdom (UK) selected from a local panel of potential respondents. Interviews, conducted by experienced qualitative research moderators, consisted of pre-decided standardised questions. Thematic analysis was undertaken, and themes were derived from the data. **Results**: A total of 26 parent interviews and 18 primary care physician interviews were conducted. Parents from all countries found the Pre-CoMiSS™ tool helpful for understanding their baby’s signs, easy to use, and useful for facilitating consultation with the physician. Physicians in the UK, Spain, and Sweden found that the Pre-CoMiSS™ tool was helpful for improving symptom reporting and for medical consultations; however, in Germany, physicians had mixed opinions, expressing concerns that the tool might increase parental anxiety, lead to overdiagnosis of CMA, and add to their workload. **Conclusions**: A parent-specific tool for recording cow’s milk-related symptoms was generally well received by parents and most physicians, though concerns about parental anxiety and workload were noted, particularly in Germany. With further validation and refinement, Pre-CoMiSS™ may be a useful tool for parents to record their infant’s symptoms related to feeding and support PCPs in considering CMA in these infants.

## 1. Introduction

Cow’s milk allergy (CMA) is an allergic reaction directed against cow’s milk proteins [1]. CMA can be immunoglobulin E (IgE) mediated, non-IgE mediated, or both [1]. Symptoms associated with severe IgE-mediated CMA include rapid onset of skin signs such as hives or urticaria; itching of the lips or mouth or erythema; angioedema; respiratory symptoms such as coughing, wheezing, or shortness of breath; vomiting; diarrhoea; or multiorgan system anaphylaxis. Late-onset symptoms such as eczema, abdominal cramps, regurgitation, vomiting, diarrhoea, colic, and blood or mucus in stools are typically associated with non-IgE-mediated CMA. Bowel dysmotility, manifesting as constipation or reflux, is considered another potential symptom of non-IgE-mediated CMA, although the topic is controversial due to limited data and the high prevalence of constipation or reflux in the general paediatric population [1,2].

Prevalence of IgE-mediated CMA is estimated at 2% of children aged 4 years or less [3,4], while the incidence of non-IgE–mediated CMA is estimated at between 0.13% and 0.72% in children under the age of 2 years, as shown in the EuroPrevall study [5]. In this population study, there was great variation between countries in identifying non-IgE-mediated allergies, indicative of poor recognition [6]. In a retrospective analysis of 1394 children aged 2 years or less in England, parent-reported hypersensitivity to cow’s milk was reported in 16.1% of children, whereas diagnosis by oral food challenge was reported in only 1.4% [7]. The clinical and financial burden associated with CMA in young children is significant, with higher prevalence of faltering growth and allergic symptoms but also of confounding infections [8,9,10,11]. This, together with the high cost of specialist formulas, increases healthcare costs and poses a financial and psychosocial burden on families [12,13].

Published guidelines or recommendations for the diagnosis and management of CMA include the World Allergy Organization Diagnosis and Rationale for Action against Cow’s Milk Allergy [14]; National Institute of Allergy and Infectious Diseases [15]; British Society for Allergy and Clinical Immunology [16]; European Society for Paediatric Gastroenterology, Hepatology and Nutrition GI Committee [17]; and Global Allergy and Asthma European Network [18] guidelines. All guidelines recommend appropriate testing when indicated, the diagnostic elimination diet, and supervised challenge or reintroduction of cow’s milk to confirm the diagnosis of CMA. However, these guidelines are not always known or followed, or the challenge test is refused by the parents [19].

There are significant challenges for both parents and physicians in diagnosing CMA. CMA symptoms, which are non-specific, overlap with common benign and self-limited disorders in infancy and other common diseases of childhood, such as gastroesophageal reflux disease and distress or crying, making diagnosis challenging [20]. As a result, patients with CMA often face a lengthy journey from first presentation of symptoms to a formal diagnosis. In the UK, a medical database study showed that it took, on average, 2.2 months from the first primary care physician visit to the first prescription of a dietary formula or milk, and 3.6 months from the first primary care physician visit to a diagnosis of CMA [21]. Additionally, in a multicentre, observational study, patients suffered from CMA-related symptoms for a mean duration of 6.9 weeks in Belgium, 9.7 weeks in Germany, 11.5 weeks in Czech Republic, and up to 24 weeks in the UK before appropriate assessment and intervention [22]. Additionally, the advised reintroduction of cow’s milk following a diagnostic elimination diet is often not performed, risking overdiagnosis of CMA and prolonged elimination diet [19]. While overdiagnosis can result in unnecessary dietary restrictions and potential micronutrient deficiencies in children, underdiagnosis leads to ongoing symptoms that may result in faltering growth and feeding difficulties at a time when appropriate nutrition for brain growth is crucial and may negatively affect patient and family quality of life [19,23,24].

The use of the internet and social media for healthcare information has grown, with 36.5% of adult allergy patients in a Cypriot study relying on it; however, these findings may not be generalisable to other populations [25]. Online symptom trackers or tools have become widely available to parents, but a recent Delphi consensus reported that many such tools were mostly non-validated and varied in their inclusion of symptoms, and were therefore not recommended by experts without the involvement of a physician [26].

The Cow’s Milk-related Symptom Score (CoMiSS™) for healthcare professionals is a widely used clinical awareness tool for CMA risk among healthcare professionals [27,28]. CoMiSS™ consists of seven questions relating to duration of crying, frequency and quantification of regurgitation, stool consistency, and skin and respiratory symptoms, in absence of an obvious cause (i.e., infection) and lasting for at least 1 week. The total score ranges from 0 to 33, and a score of ≥10 suggests the infant may benefit from an assessment for possible CMA [28,29]. To confirm the presence of CMA, diagnostic elimination, associated with resolution of or improvement in symptoms, followed by a food challenge, is necessary [11]. Although it has been shown that parents are able to reliably complete the CoMiSS™, with similar scores between paediatricians and parents [30], the tool was not intended to be used by parents. The objective of this qualitative market research study was to assess the need for a parent-reported CoMiSS™ tool (Pre-CoMiSS™) and explore its perception and potential usefulness for both parents and primary healthcare professionals.

## 2. Materials and Methods

The process of the development of Pre-CoMiSS™ is shown in Figure 1. The tool is available at the following link: https://www.cowsmilkallergy.com/precomiss (accessed on 27 April 2025).

Participants were selected from a local panel of potential respondents interested in taking part in qualitative research. The local panels, generated by Ipsos (www.ipsos.com) and its local partners, are continuously refreshed and enriched via various means, such as social media reach or local newspaper announcements. The source of respondents was independent from any of the Pre-CoMiSS™ developers. Potential respondents were then recruited by phone, email or social media, depending on the country. All went through final screening and scheduling via a telephone call. In each country, one to three respondents (three in Germany, two in the UK, two in Sweden, and one in Spain) who qualified through the screening process refused to sign the data release agreement, and so their participation in the study could not be accepted.

Parents of babies aged 2–12 months were included if infant formula had been introduced in the previous 2 months or more recently (mix breastfeeding and formula or formula only), the baby was experiencing feeding-related symptoms (such as crying, regurgitation, loose or watery stools, or skin irritation), and parents were concerned that the child might suffer from CMA due to these symptoms. The infants had no diagnosed allergy or other underlying disease. Exclusively breastfed babies were not included since CMA is uncommon in these babies, and a maternal elimination diet should only be used where there is a strong suspicion of cow’s milk being a culprit food under the guidance of a physician [31].

Recruitment of primary care physicians involved reaching out to individuals listed in market research panels, professional directories, or databases. Primary care physicians were selected if they regularly saw children aged 2–12 months for whom CMA might be suspected and if they had at least 5 years’ experience in the field. Ipsos and its local recruiters ensured a balance in gender and years of experience. Once sufficient physicians meeting the criteria were recruited, the recruitment process was closed to additional potential participants. Primary care physicians were general practitioners in the UK, Spain, and Sweden and paediatricians in Germany.

There were four female interviewers, one per country, with the local language being the mother tongue. All interviewers were experienced qualitative research moderators who had worked in the field for 15 years or more. The moderators were not related to Nestlé, other infant food industry companies, or the experts of the CoMISS™ tool. Participants were informed about the moderator’s first name and that they were working in the field of qualitative research on a project for Nestlé Health Science via Ipsos. Additionally, participants were advised that the research topic was baby feeding and were asked questions about CMA using a screening questionnaire. The participants received a monetary incentive of EUR 60 to 200 for their participation, depending on the country and whether they were parents or physicians.

Sixty-minute Zoom interviews with parents and 45 min Zoom interviews with physicians were conducted, using pre-decided standardised questions and facilitated by dedicated discussion guides. Audio recordings of interviews were undertaken to facilitate the generation of field notes. Field notes were taken during and after the interview. Live notes were taken by the moderator and observer. The transcripts were read following the interview, and notes were updated by listening back to recordings. The transcripts were not returned to the participants for comment and/or correction.

CoMiSS™ experts and Nestlé Health Science representatives listened in on some of the interviews. Their presence was communicated to participants as “other people working on the project”. Nestlé Health Science clients had their cameras and microphones off throughout the interview. All participants in the call used either first name only or initials. A sim-translator was also used in the interviews, which involved live translation of the live interviews. No post-verification of the sim-translation was performed.

Thematic analysis of the data was undertaken, with themes being derived from the data. The analysis involved a localised analysis of the interviews and a summary of key insights being provided by the local moderators. These summaries were then integrated into a broader cross-market analysis conducted by Ipsos. Ipsos attended the interviews live, with simultaneous interpretation in non-English-speaking countries, and were debriefed with local moderators. Trustworthiness was ensured using peer debriefing to revise and consolidate the findings, and the calibration of insights across different countries to validate observed patterns and assure coherence in interpretations. No software was used.

## 3. Results

### 3.1. Interviews, Participants, and Themes

A total of 26 parents and 18 primary care physicians were interviewed across four countries: Germany, Sweden, Spain, and the UK. Details of the participants are shown in Table 1 and Table 2. Participants had no connections or relations with each other. A total of five parent themes and three primary care physician themes emerged from the data (Figure 2).

### 3.2. Parent Themes

An overview of the results of the parent themes is shown in Table 3.

#### 3.2.1. Theme 1: Infant Feeding Routine and History

In all countries, parents reported a similar process of observing mild symptoms following the introduction of cow’s milk-based formula, tracking progress and then seeking advice (Table 3). Advice was sought from relatives or friends, including those who had their own children or those with medical knowledge. In Spain and Sweden, parents also reported relying on their own experiences from previous children. Symptoms reported by parents included regurgitation and spitting up of feed; eczema; dry and scaly skin on the legs, back, belly, and arms; a red rash and spotty skin around the mouth and neck; watery stools; increased stool amount and frequency; and the baby being unsatisfied and crying.


*“Once we added [infant formula], we noticed reflux, spitting and changes in poop but we were told to wait it out as this is part of changing phase”.*
(Parent, male, Germany, child’s age: 6 months)


*“We noticed a few changes when we started her on formula so I did a bit of Googling and you end up going down a wormhole!”*
(Parent, male, UK, child’s age: 4 months)


*“When I realized that my son was not gaining weight, I went to my local pharmacy to confirm it. I knew there was something wrong with his food”.*
(Parent, female, Spain, child’s age: 3 months)


*“I was more worried with my first kid. First kid regurgitated for up to 6 months. Now second born is doing the same thing. First born felt he was regurgitating as much as he was eating, but in fact he was gaining weight”.*
(Parent, male, Sweden, child’s age: 2 months)

#### 3.2.2. Theme 2: Actions Taken Following Awareness of Symptoms in Their Child

Parents from all countries reported speaking to their primary care physician at their next scheduled visit if symptoms persisted (Table 3). In the UK and Germany, most parents (four out of six and seven out of eight, respectively) did not report experiencing symptoms disturbing enough to cause the baby significant distress and to need an appointment with their physician, and parents usually waited for the next scheduled visit. Those who approached their physician found them to be mostly supportive and well informed. A few parents called their midwife or general practitioner to get advice immediately. None went to the emergency room or any other emergency service. In Spain, more educated parents often demanded to know if their baby’s symptoms required professional intervention faster than other socioeconomic classes, according to primary care physicians.


*“The paediatric nurse specialist told me babies have an immature digestive system when they’re born so I was reluctant to change brands too much to give him time to get used to things. I’m glad we waited, because it did resolve itself”.*
(Parent, female, UK, child’s age: 7 months)


*“My paediatrician just said to try this formula. Why not, I said. I don’t know any other milk”.*
(Parent, female, Spain, child’s age: 5 months)

*“I thought he would regurgitate less with this formula, but he regurgitates both breast milk and formula milk. Just something about the stomach. Sometimes, I think I could just go back [to him] taking cow’s milk again because I don’t think it has an influence. But I’m gonna wait till the allergy test”*.(Parent, female, Sweden, child’s age: 3 months)


*“Yeah, some people say, hey, it’s normal, it’s normal, but how normal is it?”*
(Parent, female, Spain, child’s age: 3 months)

#### 3.2.3. Theme 3: Concerns with Symptoms Related to Feeding

The level of awareness for CMA and attitudes varied among parents (Table 3 and Figure 3). Awareness of CMA was high in the UK, with CMA being a highly prevalent topic of discussion and the National Health Service (NHS) website directing them to possible CMA after a search of symptoms like eczema and spitting up; on the other hand, awareness was low in Germany and Sweden. In Germany, more broad information about milk intolerance rather than CMA was available to parents, and most used the term “lactose intolerance” rather than CMA. Awareness of CMA was reported to be increasing in Spain, and parents were often thought to become more “interventionist” and impatient for physicians to investigate symptoms with greater awareness.


*“CMA is one of those things that everybody on my Facebook Mums group has at some point”.*
(Parent, female, UK, child’s age: 7 months)


*“The body will get used to it and with time it will get better (…) We did not talk about it to Doc as we did not have an appt with her and this is normal baby reaction to something new”.*
(Parent, male, Germany, child’s age: 6 months)

*“I thought he would regurgitate less with this formula, but he regurgitates both breast milk and formula milk. Just something about the stomach. Sometimes, I think I could just go back [to him] taking cow’s milk again because I don’t think it has an influence. But I’m going to wait till the allergy test”*.(Parent, female, Sweden, child’s age: 3 months)

*“I do have a couple of cases of problems with milk formulas in my group of friends”*.(Parent, male, Spain, child’s age: 3 months)

#### 3.2.4. Theme 4: Sources of Information

Varied sources of information were used by parents (Table 3). Authority websites were important sources of information in the UK and Sweden, as these were deemed reliable and trustworthy. Parents in Germany tended to reach out to healthcare professionals when needing reassurance or advice. In the UK, in-person expert advice from health visitors was also important to reassure anxious parents and encourage them to visit a physician, if necessary. Online resources were widely used in the UK, Germany, and Spain. In Sweden, few parents reported using relevant forums, as these were deemed less reliable and included other parents’ beliefs and recommendations. However, Swedish parents did report undertaking a Google search of relevant words, such as “baby constipation”, “baby diarrhoea”, “baby crying”, and “baby rash”.

*“The NHS website is something that is checked, familiar and simple–it showed us the symptoms of intolerances”*. (Parent, male, UK, child’s age: 4 months)


*“I also check the brochure that we got at the hospital or at the gynaecologist. On Instagram I follow die_togs to take information and see other people[s’] experiences, etc., and I use an app called Oje, ich wachse!”*
(Parent, female, Germany, child’s age: 2 months)

*“1177 is a page where I search for information. It is reliable! There is also Familjeliv, but it[’s] more parents talking, not experts”*.(Parent, male, Sweden, child’s age: 2 months)

#### 3.2.5. Theme 5: Initial Impressions and Reactions to the Proposed Concept of Pre-CoMiSS™

The Pre-CoMiSS™ concept was very positively received by parents in all countries (Table 3 and Figure 4). The tool was clearly understood by parents as a means to record information, rather than a diagnostic tool. According to parents, the benefits of the tool included the feeling of being well informed prior to visiting the physician so their concerns would be taken more seriously. Understanding what is normal was a powerful aspect of the tool; however, parents felt that this should be highlighted more prominently. The tool was also perceived to help save time since a succinct summary of symptoms is generated. Brand fit was reported as intermediate since parents were largely positive about the involvement of Nestlé in supporting the development of the tool.

*“It’s obviously not diagnostic, it helps clarify if it could be CMA and helps you gather information to go to see your GP”*.(Parent, female, UK, child’s age: 7 months)

*“This tool just gives you something right away, it helps me during the conversation with the doctor. So you are part of that, you need to be more self-determined in what you want examined”*.(Parent, female, Germany, child’s age: 11 months)


*“Having these results will help me with the next visit to the doctor; I sometimes forget dates and times and I have to keep a diary”.*
(Parent, female, Sweden, child’s age: 5 months)


*“You have read my mind! This tool is exactly what we have been talking about before (in introduction)!”*
(Parent, female, Spain, child’s age: 5 months)

### 3.3. Primary Care Physician Themes

An overview of the results of the primary care physician themes are shown in Table 4.

#### 3.3.1. Theme 1: Experience with CMA Diagnosis

Most physicians were very aware of CMA and confident in their knowledge and ability to identify and treat CMA (Table 4). While symptoms of CMA are broad, weight gain or loss and the baby’s temperament are important factors considered by all physicians. A rate of healthy weight gain is paramount for development, and faltering growth is an important factor to be considered in diagnosis. The priority for many physicians was to alleviate symptoms while also trying to achieve a diagnosis. Recommendations or actions taken by the physician included a maternal elimination diet while breastfeeding, keeping a diary of symptoms, and prescribing a specialised formula for the management of CMA where required. Follow-up and referral to a paediatrician were further actions taken. In Germany, paediatricians are the primary care physicians; therefore, no referral was needed since they can prescribe specialised formula and perform allergy testing themselves. The reintroduction of cow’s milk to confirm the diagnosis was not discussed or recommended by physicians.


*“I don’t feel comfortable diagnosing it by myself without a second opinion–we always refer on to Pediatrics in my practice”.*
(Physician, UK)


*“Severe symptoms can be seen almost straight away but less severe ones can take some time to diagnose”.*
(Physician, Spain)

*“The major symptom I look at when I have a baby suffering is the weight of the baby and their skin condition”*.(Physician, Germany)

#### 3.3.2. Theme 2: Awareness of and Interest in CoMiSS™

Awareness of CoMiSS™ was high in the UK and Spain and low in Germany and Sweden. Interest in CoMiSS™ was high in the UK, Spain, and Sweden but low in Germany (Table 4 and Figure 5). Physicians in the UK who were unaware of CoMiSS™ prior to the interview considered it a potentially useful tool. One German physician felt encouraged to use CoMiSS™ in the future to enable more efficient and accurate identification of CMA cases.

*“We use CoMiSS with parents in our surgery (practice); I go through the questions with them, it backs up my decision-making and reassures parents. It can help set their mind at rest that it isn’t CMA”*.(Physician, UK)


*“I don’t use the score sheet anymore because I know all the symptoms babies could have related to milk”.*
(Physician, Germany)

*“I knew about CoMiSS because they presented it in a medical congress I attended some time ago”*.(Physician, Spain)


*“This looks like it’s quick and you could easily score this and get it together. This could be helpful for me myself in my own evaluation, but also for the parents…we tell them: these are the things that we look at and we end up here”.*
(Physician, Sweden)

#### 3.3.3. Theme 3: Initial Impressions and Reactions to the Proposed Concept of Pre-CoMiSS™

The concept was very positively received by physicians in the UK, Spain, and Sweden, but German physicians had some concerns that Pre-CoMiSS™ might increase parental anxiety, lead to overdiagnosis of CMA, and add to their workload (Table 4 and Figure 6).


*“This will allow parents to validate their concerns in a constructive way”.*
(Physician, UK)


*“If I could give it to some parents…but distributing to all kinds of parents, this is not a good idea. Parents feel insecure, you will create fear. Then it will steal your time if diagnosis is not clear, the parent come to you and present it with the tool, then you must take the blood test. It is taking blood in the vein, it is invasive”.*
(Physician, Germany)

*“We only have 15 min per child, so if the child has a problem and the parents have written it down, I don’t need to ask, I could read through; it could be a help even for me”*.(Physician, Sweden)

*“With this [tool] we can go straight to the point, and I will not have to ask so many questions”*.(Physician, Spain)

### 3.4. Executional Learnings and Optimisation

Participants felt that the overall concept was clear and worked well, with questions being straightforward and easy to answer. Information on “what is normal” was perhaps one of the most powerful, useful, and distinctive aspects of the concept. This information was reported as highly reassuring and fits well with the empathetic tone of the concept and should, therefore, be a key focus for any communication relating to the tool. In terms of optimisation, participants felt it would be beneficial to refer to the tool as a “diary” or “tracker”, possibly in application format, to support relevance, user experience, and credibility. Ideally, the tool should take 15 min to complete the first day, then 5 min per day on consecutive days.

Physicians reported that weight gain or loss is a key factor when considering a CMA diagnosis and is under consideration by the expert panel for Pre-CoMiSS™ tool optimisation. Additionally, capturing the duration of crying only is a potential limitation of the tool; physicians expressed that it may be more beneficial to capture the timing of crying in relation to feeding. Images, for example, of stool samples, were found to be very useful, reassured parents about what is “normal”, and made questions easier to answer. Additional photos, such as skin symptoms among ethnic skin variations, would help direct answers, which would be beneficial to users. Audios of different respiratory sounds would also be beneficial for identification of respiratory symptoms associated with CMA. The opportunity of voice-activated completion or short-form answers with tick boxes to make it easier to complete while feeding or soothing a baby would also be beneficial to users.

## 4. Discussion

The study explored how parents and physicians in four European countries (Germany, Sweden, Spain, and the UK) perceived the Pre-CoMiSS™ tool for tracking cow’s milk-related symptoms. Parents found the tool helpful, easy to use, and useful for discussing symptoms with physicians. Physicians in the UK, Spain, and Sweden viewed the tool positively, as it could improve symptom reporting and medical consultations. However, German paediatricians had mixed opinions, expressing concerns that the tool might increase parental anxiety, lead to overdiagnosis of CMA, and add to their workload. Differences between countries were also observed. Parents in the UK and Spain were more aware of CMA and more likely to suspect it in their child, while German parents tended to see symptoms as part of normal development. In Sweden, CMA awareness among parents was low. These findings suggest that Pre-CoMiSS™ could help parents document symptoms, but its implementation should consider cultural and clinical differences to ensure its usefulness in different healthcare settings.

This qualitative study provides high-quality, in-depth feedback concerning parents’ and physicians’ perspectives of the Pre-CoMiSS™ tool, and contributes significantly to existing research on CMA awareness and symptom tracking. Moreover, it has enabled the exploration of cultural differences via its multi-country design, and its practical relevance highlights real-world applications. The study included a dual perspective of the Pre-CoMiSS™ tool from both parents and physicians; taking both viewpoints into consideration is important for successful implementation. In addition, it offers the opportunity for tool optimisation and is the foundation for future research to validate Pre-CoMiSS™ in clinical settings.

The small sample size of the study limits the generalisability of findings; however, the sample reported here is similar to that of many qualitative studies, and the depth of evidence provided in this study with regards to parent and physician perspectives of the Pre-CoMiSS™ tool is greater than what could be provided in a quantitative study with a larger sample size. Furthermore, the study lacks statistical validation and longitudinal data due to its qualitative design. Additionally, the qualitative design of the study means that results are descriptive, which may limit conclusions that can be drawn as well as the generalisability of the findings. Nevertheless, the focus of the study is on participant perspectives, which were captured in detail using this type of methodology. Since participants were recruited via local panels from Ipsos and its local partners, there was potential for selection bias. However, as previously mentioned, these local panels are continuously refreshed and enriched via various means to enhance the representativeness of the panel. The use of unverified simultaneous translation is a further potential limitation; however, live interpretation was performed by reliable and experienced professional simultaneous interpreters who were briefed about the project and tested material, and the analysis was not based on simultaneous interpretation alone. The study was also restricted to four European countries, which may limit the generalisability of the results to other countries within and outside of Europe. However, the cultural differences between these countries have assisted in gathering a range of opinions and perspectives. Additionally, the study was supported by Nestlé Health Science, and their medical and scientific committee were able to listen in on some of the interviews. Nevertheless, the study was fully independent, and Nestlé Health Science did not have any impact on the design, execution, or outcomes reported. Further limitations include the lack of data on the socioeconomic or educational level of participants, and the high prevalence of female participants and primiparous mothers, which have the potential to influence findings. Differences in physician specialities as well as variability in physicians’ awareness of CoMiSS™ could also have influenced the findings. Finally, the tool has not been tested in real clinical practice; validation in this setting is required.

Parent knowledge and awareness of food allergies has previously been reported to be poor, with one parent survey reporting Food Allergy Knowledge Test scores of 30 out of 59 for parents of children with a food allergy diagnosed via a paediatrician or allergist and a score of 26 for parents with children without a food allergy [32]. Higher Food Allergy Knowledge Test scores indicate greater knowledge; however, a cutoff score for an adequate level of knowledge has not yet been established [32]. Additionally, a cross-sectional study of parents attending a primary care health centre in the Palestinian Authority showed low numbers of correct responses regarding treatment of food allergies (18.6% to 24.9%) [33]. Parent education level, income, and having a child with a known food allergy were all found to be associated with greater awareness and knowledge of food allergy, while parent anxiety and ethnic minority were associated with lower knowledge [32,33]. Attitudes of parents of children with CMA have been showed to be heavily influenced by emotions, and this can affect parental understanding and communication of guideline recommendations [34].

Use of online sources of information for researching a variety of healthcare issues is becoming increasingly common [25]. In a Spanish university hospital, 53% of allergy patients or parents of paediatric allergy patients used the internet to search for information prior to their allergy clinic visit and 47% after their visit [35]. The most common sources of information were blogs, forums, or testimonials, which were used by 55% of patients to gain additional in-depth knowledge of their condition [35]. A further cross-sectional study from Spain demonstrated that 30.3% of patients or parents of patients with food allergies who used social media used it for food allergy-related searches, such as food safety information [36]. A systematic review has shown that applications used for food allergy purposes, including meal planners, food product information, and restaurant applications, have overall acceptable quality, but further testing in clinical trials is required [37]. In a cross-sectional study conducted in the Palestinian Authority, online sources of information were frequently reported among parents, with 65.4% using online sources compared with 37.8% using healthcare services and 7.1% using support groups [33]. In contrast, most parents of children with allergies from Cyprus reported the paediatrician as their first-line source of information (44.5%), with only 6.2% first searching online for information [25]. In a further survey of Turkish parents of a child with CMA, 75% of parents considered online sources of information prior to visiting their physician; however, many parents reported that online information contradicted information provided by their physician [38]. Furthermore, in a qualitative study of parent preferences for information pertaining to food allergies, parents declared having insufficient information and experiencing frequent instances where their questions or concerns were not answered sufficiently [39]. Since information from online resources can be outdated or misleading, the physician has a vital role to play in providing up-to-date clinical information to their patients as well as discussing and addressing concerns regarding any information obtained from online sources [40].

Physician awareness of CMA and experience with CMA diagnosis is generally high [41,42]; however, some physicians have reported an interest in additional training within this field [42]. A survey of paediatricians and paediatric residents from Turkey revealed an adequate level of knowledge regarding CMA, with a score of 7.5 and 8.3 (out of 10), respectively [43]. Nevertheless, additional academic training led to an increase in score to 10.0 and 9.7 for the paediatric residents and paediatricians, respectively [38]. Knowledge gaps in the diagnosis and management of CMA have previously been highlighted; these include differences in IgE- versus non-IgE-mediated CMA, as well as the similarity in symptoms between CMA and lactose intolerance [42]. Clinical management of CMA, specifically in terms of formulas used, has been found to differ among physicians and guideline recommendations [36,37]. Furthermore, awareness of CMA guidelines by general practitioners in the UK was reported to be low, with only 5% being very familiar with the UK CMA guidelines [44]. Additional physician training in the management and treatment of CMA is important to increase awareness of CMA and guideline recommendations, as well as to close any knowledge gaps [43].

CoMiSS™ has been shown to be an important awareness tool for CMA, with a score of ˂6 being highly predictive of the absence of CMA and a score of ≥10 predicting the possible risk of CMA [24]. A recent Delphi consensus on the use of online symptom questionnaires for CMA highlighted that CoMiSS™ is the only validated online symptom questionnaire [26]. Additionally, real-world data suggest that physician satisfaction with CoMiSS™ is high [24]. Although it has been recommended that such tools should not be used without a physician’s input [26], a recent study comparing the use of CoMiSS™ by parents and physicians demonstrated high reliability of parent scoring, highlighting the potential usefulness of a parent-specific symptom score [30].

Concerns about patient anxiety and additional workload have previously been reported by German general practitioners, with 77% of 2532 physicians participating in a survey stating that the use of online health information was a major challenge faced in their day-to-day practice, affecting patients’ mental state as well as the request for instrumental diagnosis [40]. Additionally, a systematic review showed high levels of anxiety in 14 to 52% of parents using online resources for health-related information [45]. However, Skranes et al. showed that regular use of a specific website containing child health information for parents, which was created and managed by physicians in Norway, led to reduced anxiety and increased knowledge among mothers [46]. A further concern was the overdiagnosis of CMA, which is relatively common, with initial concern raised in more than 20% of parents of children under 2 years of age with symptoms of potential CMA [7]. This can lead to unnecessary use of specialised infant formulas and dietary elimination [47,48]. These concerns could potentially be mitigated by physicians having control over the distribution of the Pre-CoMiSS™ tool.

Differing views of CMA diagnosis and tools such as CoMiSS™ and Pre-CoMiSS™ by the physicians within the study are likely a result of the contrasting healthcare systems in the four European countries. UK physicians may wish to resolve CMA symptoms quickly to reduce the burden on the national healthcare system, and online resources may help to reduce the demand for appointments at general practice surgeries [21,49]. Spain and Sweden have similar healthcare systems to the UK. In contrast, the German healthcare system operates a health insurance model. The fact that CMA diagnosis is not seen as a priority, provided the child is thriving, is reflected in the low levels of diagnostic skin prick and IgE testing requested by German physicians [22]. Additionally, a survey of German medical faculty members found heterogeneity in their perspectives on digitisation in the healthcare sector, with many stating that the concept of using digital assistance in core capabilities such as diagnosis was difficult for doctors [50].

Following these positive first impressions of Pre-CoMiSS™, future steps will involve additional validation and refinement of the tool. Capturing height and weight changes and the timing of crying in relation to feeding are under consideration by the expert panel for Pre-CoMiSS™ tool optimisation. Additionally, assessing the impact of the tool on overdiagnosis of CMA and parental anxiety is needed. Furthermore, capturing Pre-CoMiSS™ scores in presumed healthy infants and comparing the scores to those of children with CMA will provide further evidence of the usefulness of the tool. In the future, ensuring broader accessibility through independent medical platforms or direct distribution by healthcare professionals may enhance its clinical adoption.

## 5. Conclusions

A parent-specific tool for recording cow’s milk-related symptoms was well received by parents and most physicians in this study. With further validation and refinement, Pre-CoMiSS™ has the potential to improve symptom documentation and enhance physician–parent communication regarding possible CMA. However, concerns from some physicians, particularly in Germany, highlight the need for careful implementation to avoid unnecessary parental anxiety and increased workload for healthcare professionals.

## Figures and Tables

**Figure 1 nutrients-17-01563-f001:**
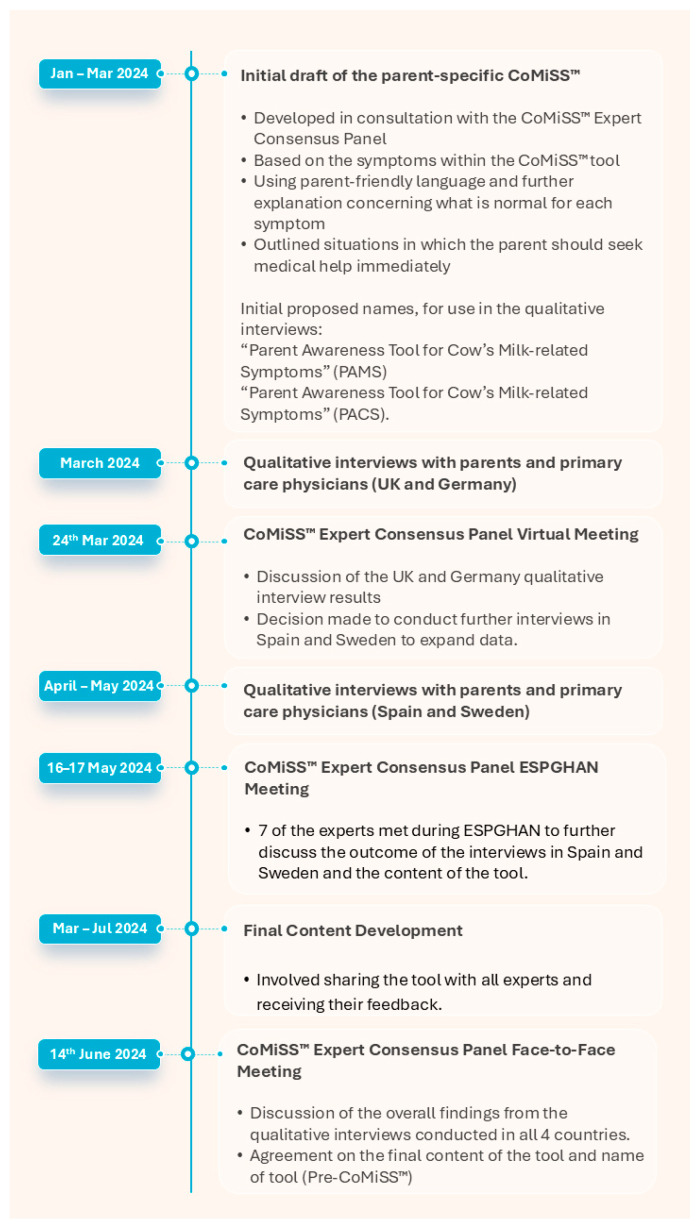
Development of Pre-CoMiSS™. CoMiSS—Cow’s Milk-related Symptom Score; ESPGHAN—European Society for Paediatric Gastroenterology, Hepatology and Nutrition; Pre-CoMiSS—parent-reported CoMiSS.

**Figure 2 nutrients-17-01563-f002:**
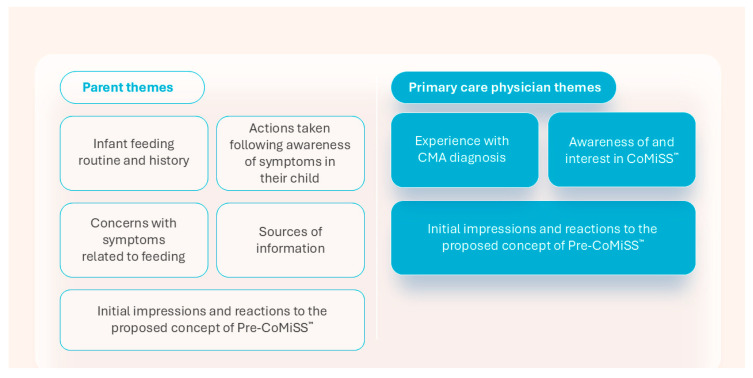
Parent and primary care physician themes. CMA—cow’s milk allergy; CoMiSS—Cow’s Milk-related Symptom Score; Pre-CoMiSS—parent-reported CoMiSS.

**Figure 3 nutrients-17-01563-f003:**
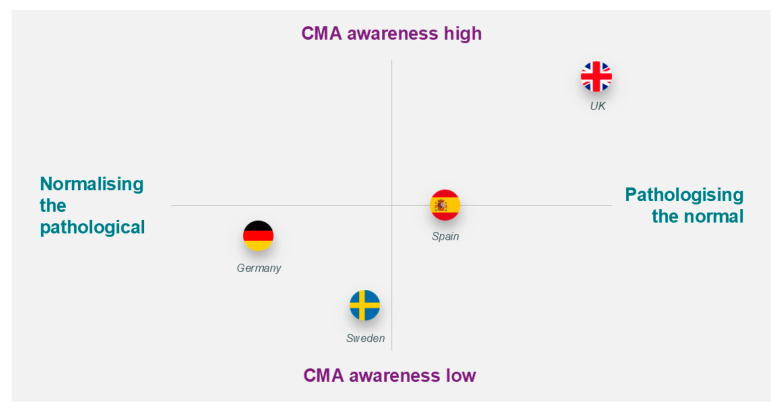
Illustration of the level of awareness and attitudes. CMA—cow’s milk allergy.

**Figure 4 nutrients-17-01563-f004:**
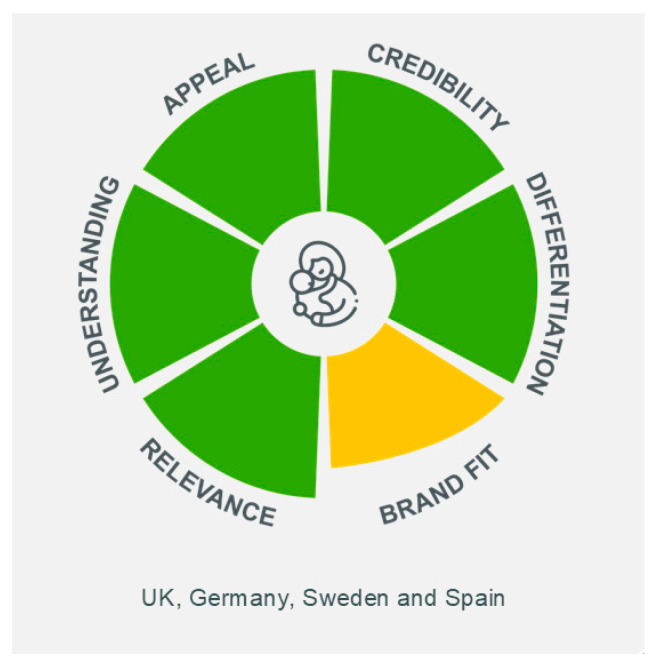
Parent key performance indicators of Pre-CoMiSS™. Green = high; yellow = intermediate; Pre-CoMiSS—parent-reported Cow’s Milk-related Symptom Score.

**Figure 5 nutrients-17-01563-f005:**
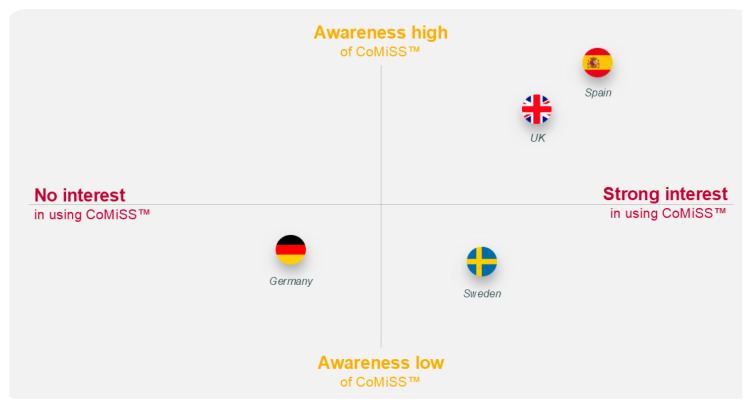
Illustration of primary care physician awareness of and interest in CoMiSS™. CoMiSS—Cow’s Milk-related Symptom Score.

**Figure 6 nutrients-17-01563-f006:**
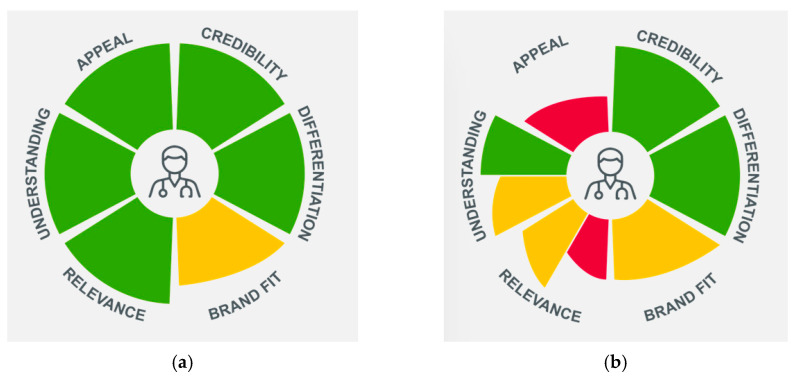
Primary care physician key performance indicators of Pre-CoMiSS™: (**a**) UK, Sweden, and Spain; (**b**) Germany. Green = high; yellow = intermediate; red = low. Note that in Germany, understanding and relevance were polarised; Pre-CoMiSS—parent-reported Cow’s Milk-related Symptom Score.

**Table 1 nutrients-17-01563-t001:** Primary care physician demographics.

Physician	Country	Gender	Years Practicing	Workplace
H1	Sweden	Male	18	Private clinic
H2	Sweden	Female	14	Healthcare centre
H3	Sweden	Female	9	Healthcare centre
H4	Sweden	Male	35	Healthcare centre
H5	UK	Female	12	Individual NHS practice
H6	UK	Female	7	Group NHS practice/community GP
H7	UK	Male	30	Individual NHS practice
H8	UK	Female	20	Group NHS practice/community GP
H9	UK	Male	16	Group NHS practice/community GP
H10	UK	Female	8	Individual NHS practice
H11	Spain	Female	13	Children’s healthcare centre
H12	Spain	Female	29	Children’s healthcare centre
H13	Spain	Male	26	Children’s healthcare centre
H14	Spain	Female	25	Children’s healthcare centre
H15	Germany	Female	19	Group private practice
H16	Germany	Female	23	Group private practice
H17	Germany	Male	9	Private practice
H18	Germany	Male	26	Private practice

GP—general practitioner; NHS—National Health Service.

**Table 2 nutrients-17-01563-t002:** Parent demographics.

Parent	Country	Gender	Age (Years)	Birth Order of Child	Child’s Age (Months)
P1	Sweden	Male	28	Second	2
P2	Sweden	Female	39	Third	10
P3	Sweden	Male	42	First	3
P4	Sweden	Female	36	Third	7
P5	Sweden	Female	25	First	3
P6	Sweden	Female	40	First	4
P7	UK	Female	29	First	7
P8	UK	Male	31	First	4
P9	UK	Female	39	Second	3
P10	UK	Female	35	First	6
P11	UK	Female	35	First	6
P12	UK	Female	41	Second	7
P13	UK	Male	34	First	5
P14	UK	Male	35	First	4
P15	Spain	Female	32	Second	10
P16	Spain	Male	37	First	3
P17	Spain	Female	38	Second	5
P18	Spain	Female	30	First	9
P19	Spain	Female	28	First	3
P20	Spain	Male	37	First	6
P21	Germany	Female	30	First	10
P22	Germany	Male	41	First	2
P23	Germany	Female	34	Second	11
P24	Germany	Female	29	Second	7
P25	Germany	Female	32	First	2
P26	Germany	Male	35	Second	6

**Table 3 nutrients-17-01563-t003:** Overview of the results of the parent themes.

Theme	Summary	Key Differences	
Infant feeding routine and history	All parents undertook a similar process of observing mild symptoms following the introduction of cow’s milk-based formula, tracking progress then seeking advice, with no immediate recourse to the physician.	UK and Germany	Sought information from Google Lack of weight gain not reported as a symptom Nervousness surrounding the introduction of infant formula
		Sweden and Spain	Sought information from specialised websites Also felt at ease contacting a non-MD healthcare professional, such as a nurse or pharmacist First-time parents were more unsure about which symptoms were normal
Actions taken following awareness of symptoms in their child	All parents reported speaking to their primary care physician at their next scheduled visit if symptoms persisted.	UK and Germany	Sought to align symptoms, e.g., eczema or spitting up, with CMA or milk intolerance Many (UK 5, Germany 4) wondered whether they should change their child’s formula, but often waited to see how symptoms developed Considered reducing or stopping own cow’s milk intake, if breastfeeding
		Sweden and Spain	Most (Sweden 4, Spain 4) sought physician’s advice before deciding if they should change to a specialised formula A few (Sweden 2, Spain 2) were open to trying anti-reflux or “sensitive” formula based on a friend’s advice or from previous experience without consulting their physician
Concerns with symptoms related to feeding	The level of awareness for CMA and attitudes varied among parents.	UK and Spain	Pathologising the normal (interpreting normal, everyday behaviours or symptoms as indicative of a medical problem) Parents more vigilant and concerned about mild symptoms in their infants Awareness was high (UK) or increasing (Spain)
		Germany and Sweden	Normalising the pathological (viewing symptoms that could indicate a medical issue as normal or insignificant) Parents perceived mild symptoms as part of normal infant development Low awareness
Sources of information	Varied sources of information were used by parents.	UK	NHS website (www.nhs.uk) Health visitors Internet forums (e.g., www.mumsnet.com) and social media
		Sweden	Authority websites (www.1177.se, www.knodd.se and www.rikshandboken-bhv.se) Little use of relevant forums (e.g., www.familjeliv.se)
		Germany	Forums or blogs (e.g., www.gutefrage.net, www.richtigwissen.de, and www.gofeminin.de) and social media Newsletters (e.g., Apotheken-Umschau) Healthcare professionals such as midwives, paediatricians, or pharmacists Experience of family members and friends with children
		Spain	Search engines, social media, physician blogs, or infant formula brand websites Experience of family members and friends with children Books by experts dedicated to newborn topics (e.g., Ser mamá by Nazareth Oliver)
Initial impressions and reactions to the proposed concept of Pre-CoMiSS™	Empathetic tone was seen as supportive, encouraging them to stay calm Informative, helping parents to be well informed prior to visiting the physician, but also approachable and realistic High credibility since it was developed by experts No similar tools for parents available Provides feelings of reassurance, control, and confidence Positive about involvement of Nestlé		

CMA—cow’s milk allergy; MD—Doctor of Medicine; NHS—National Health Service; Pre-CoMiSS—parent-reported Cow’s Milk-related Symptom Score.

**Table 4 nutrients-17-01563-t004:** Overview of the results of the physician themes.

Theme	Summary	Key Differences	
Experience with CMA diagnosis	Most physicians were very aware of CMA and confident in their knowledge and ability to identify and treat CMA. All physicians reported seeing many suspected or mild cases, but most had little or no experience with severe CMA cases. The diagnostic protocol for CMA was largely the same in all countries, with a detailed history being routinely taken to rule out other conditions.	UK	Some preferred to, or protocol dictated that they, refer cases to a paediatrician High concern about underdiagnosis Tended to rule out less severe conditions, such as reflux and colic Challenges: time-consuming nature of making a diagnosis, especially if referral was required, and the complexity of diagnosis
		Germany	If the baby continued to thrive, diagnosing CMA was not a priority Tended to rule out more severe conditions, e.g., thyroid conditions
		Sweden	Keen to speed up CMA diagnosis but not too concerned about missing cases Tended to rule out more severe conditions, e.g., thyroid conditions Challenges: time-consuming nature of making a diagnosis, especially for milder cases
		Spain	Keen to speed up CMA diagnosis but were not too concerned about missing cases Tended to rule out more severe conditions, e.g., thyroid conditions Challenges: time-consuming nature of making a diagnosis, especially for milder cases Greater parental awareness causes greater scrutiny of symptoms and increased pressure to diagnose by parents
Awareness of and interest in CoMiSS™	Awareness of CoMiSS™ was high in the UK and Spain and low in Germany and Sweden. Interest in CoMiSS™ was high in the UK, Spain, and Sweden but low in Germany.	UK	Approximately half of the GPs were familiar with CoMiSS™ Most used it as a framework for diagnosis during appointments One shared it directly with parents to complete for themselves
		Germany	Most saw no need for using CoMiSS™ since they felt experienced, competent, and comfortable enough to judge CMA symptoms without it
		Sweden	Awareness of CoMiSS™ was non-existent High interest; potential use as a checklist to make sure that no symptoms have been forgotten
		Spain	Three were familiar with CoMiSS™ Two actively used it, integrating it with their usual diagnostic approach to create a more complete picture
Initial impressions and reactions to the proposed concept of Pre-CoMiSS™	The concept was very positively received by physicians in the UK, Spain, and Sweden, but German physicians had some concerns.	UK, Sweden and Spain	Very positively received Appealing and relevant, with an empathetic tone and no issues in understanding
		Germany	Not needed Concerns: ○ Potential to pathologise normal, everyday symptoms that can occur frequently in babies, leading to parental anxiety ○ Increased workload due to the need to check printed data, which parents might bring with them to appointments, and the need to reassure parents ○ Potential to lead to parents putting increased pressure on physicians to diagnose CMA using invasive and often unreliable diagnostic methods However, two physicians recognised the benefit for parents

CMA—cow’s milk allergy; CoMiSS—Cow’s Milk-related Symptom Score; GP—general practitioner; Pre-CoMiSS—parent-reported CoMiSS.

## Data Availability

The original contributions presented in this study are included in the article. Further inquiries can be directed to the corresponding author.

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
