# Peer review of "Evaluating the Need for Pre-CoMiSS™, a Parent-Specific Cow’s Milk-Related Symptom Score: A Qualitative Study"

_nutrients, 2025, doi:10.3390/nu17091563_

Round 1
Reviewer 1 Report
Comments and Suggestions for Authors
This qualitative study provides valuable initial insights into the perceived need and potential utility of a parent-reported symptom score (Pre-CoMiSS™) for infants with suspected Cow's Milk Allergy (CMA) across four European countries. The dual perspective from both parents and physicians is a strength, highlighting key areas of concordance and divergence, particularly regarding the tool's benefits and potential drawbacks (e.g., parental anxiety, physician workload). The findings offer a useful foundation for the further development and validation of the Pre-CoMiSS™ tool. However, significant revisions are required to address several critical concerns before the manuscript can be considered further for publication.
- Provide robust justification for the "IRB Statement: Not applicable" or obtain and document appropriate ethical approval/waiver. This is a critical point requiring immediate clarification.
- Address the significant limitation posed by the use of unverified simultaneous translation and its potential impact on data accuracy.
- Enhance the description of the thematic analysis process to demonstrate sufficient rigor (e.g., detailing coding procedures, ensuring trustworthiness).
- More thoroughly discuss the potential for selection bias arising from recruitment via pre-existing panels.
- Acknowledge and discuss the potential influence of sponsor representatives observing interviews as a study limitation.
- Deepen the discussion regarding physician concerns (parental anxiety, overdiagnosis, workload) and propose mitigation strategies for tool development and implementation.
- Expand the limitations section to explicitly incorporate the concerns regarding ethical oversight, translation methodology, potential selection bias, and observer effects.
- Ensure the limitations regarding geographical scope (four European countries) and its impact on generalizability are clearly stated.
Author Response
Thank you very much for taking the time to review the manuscript entitled Evaluating the need for Pre-CoMiSS™, a parent-specific Cow's Milk-related Symptom Score: A qualitative study (manuscript ID nutrients-3597454). Please find below the detailed responses to each of your comments and the corresponding revisions are highlighted in the re-submitted manuscript files.
Comment 1: Provide robust justification for the "IRB Statement: Not applicable" or obtain and document appropriate ethical approval/waiver. This is a critical point requiring immediate clarification. |
Response 1: Thank you for your comment. We believe that ethical approval is not needed for this study since it is a qualitative market research study, obtaining parent and doctors opinions on the Pre-CoMiSS™ tool. The research did not involve medical procedures, treatments, or interventions that would necessitate an ethics review and no data was obtained from healthcare records. According to the United Kingdom Department of Health Governance arrangements for Research Ethics Committees document (https://s3.eu-west-2.amazonaws.com/www.hra.nhs.uk/media/documents/GAfREC_Final_v2.1_July_2021_Final.pdf), statement 2.3.15 “Market research may be undertaken by professional market researchers, e.g. for public health research or on behalf of pharmaceutical or medical device companies. Where such research is conducted by professional market researchers in accordance with the principles set out in the Market Research Society Code of Conduct or with the Legal and Ethical Guidelines issued by the British Healthcare Business Intelligence Association (BHBIA), it does not require REC review”. In this study, respondents signed a standard market research agreement waiver and confidentiality agreement, in line with General Data Protection Regulation regulations. They gave explicit consent for recording at the beginning of the interview, before the recording began. The study complied with the ICC/ESOMAR International Code on Market, Opinion and Social Research and Data Analytics, which provides guidelines for conducting ethical research. |
Comment 2: Address the significant limitation posed by the use of unverified simultaneous translation and its potential impact on data accuracy. |
Response 2: Thank you for your comment. We acknowledge the limitation of using unverified simultaneous translation and have included this in the limitations section of the discussion (page 16, lines 466-469). However, the live interpretation was performed by reliable professional simultaneous interpreters who are experienced and accustomed to this kind of work and who were briefed about the project and tested material. Additionally, the analysis was not solely based on simultaneous interpretation. Local moderators provided a localized analysis of the interviews and reviewed the global report for any inaccuracies, ensuring content was corroborated by local speakers. In addition to the local-level analysis, Ipsos Switzerland cross-referenced the local summaries to ensure consistency and accuracy. Hence, while the use of unverified simultaneous interpretation is a limitation, the measures taken significantly mitigated its impact. |
Comment 3: Enhance the description of the thematic analysis process to demonstrate sufficient rigor (e.g., detailing coding procedures, ensuring trustworthiness) |
Response 3: Thank you for your comment. We agree that the details of the thematic analysis should be further outlined. We have therefore updated this within the methods section (page 5, lines 185-192). The analysis process was not based on a coding approach. The analysis was conducted at the local level by experienced moderators who conducted the interviews and summarized key insights. These summaries were then integrated into a broader cross-market analysis done by Ipsos. Ipsos attended interviews live (with sim-interpretation for non-English speaking countries) and debriefed with local moderators. The local moderators were pivotal in reflecting on respondent discussions. They captured nuanced insights on themes such as 'awareness of CMPA' from their interviews, using live notes and AI-generated transcripts as a resource. Ipsos conducted a cross-market synthesis, integrating insights from different locales to identify overarching themes and patterns. Trustworthiness was ensured through the expertise of the moderators and a systematic approach of cross-referencing local summaries to maintain consistency and robustness in capturing diverse perspectives. Specific measures to ensure trustworthiness also included peer debriefing to revise and consolidate findings, and the calibration of insights across different markets to validate observed patterns and assure coherence in interpretations. Additionally, key insights are supported by respondent verbatim, providing direct quotes from participants, which further enhances the credibility and trustworthiness of the research findings. |
Comment 4: More thoroughly discuss the potential for selection bias arising from recruitment via pre-existing panels. |
Response 4: Thank you for your comment. The recruitment process used in this study was conducted as such since it was a market research study. Ipsos is a market research company that assisted in this initiative. We agree that there is the possibility of selection bias and have expanded on this point with the discussion section of the manuscript (page 16, lines 464-466). |
Comment 5: Acknowledge and discuss the potential influence of sponsor representatives observing interviews as a study limitation. |
Response 5: Thank you for your comment. The involvement of sponsor representatives has already been acknowledged and discussed within the limitations section of the discussion (page 16, lines 472-475). |
Comment 6: Deepen the discussion regarding physician concerns (parental anxiety, overdiagnosis, workload) and propose mitigation strategies for tool development and implementation. |
Response 6: Thank you for your comment. We agree that there is the need for more discussion regarding these physician concerns. We have therefore included some additional text within the discussion section of the manuscript, to include mitigation strategies (page 17, lines 545-558). |
Comment 7: Expand the limitations section to explicitly incorporate the concerns regarding ethical oversight, translation methodology, potential selection bias, and observer effects. |
Response 7: Thank you for your comment. We have updated the limitations section of the manuscript to incorporate the limitations concerning the simultaneous translation (page 16, lines 466-469), potential selection bias (page 16, lines 464-466) and observer effects (page 16, lines 472-475). Please refer to response 1 with regards to ethics approval. |
Comment 8: Ensure the limitations regarding geographical scope (four European countries) and its impact on generalizability are clearly stated. |
Response 8: Thank you for your comment. We agree that the geographical scope may be seen as somewhat limited and this may impact the generalizability of the results. We have therefore included this within the discussion section of the manuscript (page 16, lines 469-472). |
Reviewer 2 Report
Comments and Suggestions for Authors
The manuscript presents an international qualitative study addressing the practical experience using a screening tool ( Cow´s Milk -related Symptom score (CoMISS) in four European countries.
In the introduction the background of Cow´s milk allergy is described meeting the pathophysiology, the prevalence, guidelines and recommendations for diagnostic and therapeutic approaches.
The aim of the study was to explore the feasibility using the CoMISS tool by parents and primary healthcare professionals.
The recruiting process was supported by a commercial platform (www.Ipsos.com) . Fig 1 describes the recruiting process properly. It is a quite unusual way providing the risk of selection bias.
The presentation of the results is comprehensive reporting on the main topics related to Cow´s milk allergy. The interpretations (Theme 5 ff) of Table 3 and 4, page 10-13 are not supported by structured data but only descriptive. This may have been the only way to face the very low number of participants in the parental group. The fact, that some physicians were related to more than one parental participant may influence the level of reliable information.
In the discussion the authors underline the limitations effected by the small sample size of the study.
The interesting finding of very different views from health professionals in the participating countries is worth to discuss more profoundly
Overall the study provides some interesting aspects relating to the use of CoMISS as a screening tool.
The descriptive design does not allow any conclusions of potential benefits. The authors should consider to concentrate on the main aspects of the report by shortening the narrative.
Author Response
Thank you very much for taking the time to review the manuscript entitled Evaluating the need for Pre-CoMiSS™, a parent-specific Cow's Milk-related Symptom Score: A qualitative study (manuscript ID nutrients-3597454). Please find below the detailed responses to each of your comments and the corresponding revisions are highlighted in the re-submitted manuscript files.
Response 1: Thank you for your comment. The recruitment process used in this study was conducted as such since it was a market research initiative. Ipsos is a market research company that assisted in this initiative. We agree that there is the possibility of selection bias and have expanded on this point with the discussion section of the manuscript (page 16, lines 462-466). |
Comments 2: The presentation of the results is comprehensive reporting on the main topics related to Cow´s milk allergy. The interpretations (Theme 5 ff) of Table 3 and 4, page 10-13 are not supported by structured data but only descriptive. This may have been the only way to face the very low number of participants in the parental group. The fact, that some physicians were related to more than one parental participant may influence the level of reliable information. |
Response 2: Thank you for your comment. The data presented in the manuscript is qualitative in nature. The depth of evidence provided in this study with regards to parent and physician perspectives of the Pre-CoMiSS™ tool is greater than what could be provided in a quantitative study with a larger sample size. We have already discussed the limited sample size within the discussion section of the manuscript (page 15, lines 454-458). The limitations and advantages of the descriptive nature of the study has been further expanded within the discussion (page 15, lines 458-461). |
Comments 3: The interesting finding of very different views from health professionals in the participating countries is worth to discuss more profoundly. |
Response 3: Thank you for your comment. We agree that the very different views of the health professionals in the participating countries is worth further discussion. We have therefore included some additional text within the discussion section of the manuscript (pages 17-18, lines 559-570). |
Comments 4: The descriptive design does not allow any conclusions of potential benefits. The authors should consider to concentrate on the main aspects of the report by shortening the narrative. |
Response 4: Thank you for your comment. The limitations and advantages of the descriptive nature of the study has been further expanded within the discussion section of the manuscript (page 15, lines 454-461). We do not believe that the narrative needs to be shortened since we the results need to be discussed thoroughly in the context of the literature. |
Round 2
Reviewer 1 Report
Comments and Suggestions for Authors
good to go